# High Atlastin 2-2 (ATL2-2) Expression Associates with Worse Prognosis in Estrogen-Receptor-Positive Breast Cancer

**DOI:** 10.3390/genes14081559

**Published:** 2023-07-29

**Authors:** Inga Reynisdottir, Adalgeir Arason, Edda S. Freysteinsdottir, Sigrun B. Kristjansdottir, Bylgja Hilmarsdottir, Gunnhildur A. Traustadottir, Oskar T. Johannsson, Bjarni A. Agnarsson, Rosa B. Barkardottir

**Affiliations:** 1Cell Biology Unit, Department of Pathology, Landspitali—The National University Hospital of Iceland, 101 Reykjavik, Iceland; 2BMC (Biomedical Center), Faculty of Medicine, University of Iceland, 101 Reykjavik, Iceland; adalgeir@landspitali.is (A.A.); bylgjahi@landspitali.is (B.H.); gunasta@landspitali.is (G.A.T.); rosa@landspitali.is (R.B.B.); 3Molecular Pathology Unit, Department of Pathology, Landspitali—The National University Hospital of Iceland, 101 Reykjavik, Iceland; eddasf@landspitali.is; 4Department of Pathology, Landspitali—The National University Hospital of Iceland, 101 Reykjavik, Iceland; sigrunk@landspitali.is (S.B.K.); bjarniaa@landspitali.is (B.A.A.); 5Department of Oncology, Landspitali—The National University Hospital of Iceland, 101 Reykjavik, Iceland; oskarjoh@landspitali.is; 6Faculty of Medicine, University of Iceland, 101 Reykjavik, Iceland

**Keywords:** Atlastin 2, ATL2, ATL2-2, breast cancer, survival, endoplasmic reticulum

## Abstract

The disruption of endoplasmic reticulum (ER) homeostasis occurs in many human diseases. Atlastins (ATLs) maintain the branched network of the ER. The dysregulation of ATL2, located at ER network junctions, has been associated with cancer. ATL2 is necessary for lipid droplet formation in murine breast tissue. Thus, we analyzed whether ATL2 has a role in human breast cancer (BC) pathology. The expression of ATL2 variant ATL2-2 was analyzed in breast tumors from the BC cohorts of the TCGA, METABRIC, and two independent Icelandic cohorts, Cohort 1 and 2; its association with clinical, pathological, survival, and cellular pathways was explored. ATL2-2 mRNA and protein expression were higher in breast tumors than in normal tissue. ATL2-2 mRNA associated with tumor characteristics that indicate a worse prognosis. In METABRIC, high ATL2-2 mRNA levels were associated with shorter BC-specific survival (BCSS) in patients with estrogen-receptor-positive luminal breast tumors, which remained significant after correction for grade and tumor size (HR 1.334, CI 1.063–1.673). Tumors with high ATL2 mRNA showed an upregulation of hallmark pathways MYC targets v1, E2F targets, and G2M checkpoint genes. Taken together, the results suggest that high levels of ATL2-2 may support BC progression through key cancer driver pathways.

## 1. Introduction

Determining the targets for blocking increased breast cancer (BC) metastasis remains a pressing need. The endoplasmic reticulum (ER) participates in the unfolded protein response (UPR), ER-associated degradation (ERAD), and ER macrophagy [1]. The UPR is an important sensor of misfolded proteins whose dysregulation induces chronic ER stress that may result in diseases, including cancer [2,3]. ER stress can affect both the tumor and the tumor microenvironment and, as such, participate in the regulation of metastasis [4].

Atlastin 1-3 (ATL1-3) are a family of membrane-bound GTPase proteins located in the ER. ATL1 is expressed everywhere in the ER, whereas ATL2 and ATL3 are mainly expressed at the ER junctions. Their main function is to join ER membrane tubules into a branched network [5,6,7]. The depletion of ATL leads to a less branched ER [5,6,8], a phenotype that can be rescued by the expression of ATL2 [9]. ATL proteins have an N-terminal GTPase domain, necessary for membrane fusion and the generation of three-way junctions, followed by a three-helical bundle domain (3HB), two membrane domains, and a short α-helical C-terminus [8,10,11]. The ATL proteins maintain the homeostasis of the ER network along with the Lunapark and Reticulon proteins [12]. ATL2 is a novel substrate for ubiquitination by Lunapark [13], which results in its degradation. Various other roles have been ascribed to the ATL proteins. They participate in the autophagic degradation of the ER membranes, ER-phagy, by restructuring the membranes to bring FAM134B (Reticulophagy Regulator 1), which binds LC3 (Microtubule-Associated Protein 1 Light Chain 3 α), into the autophagosome [1]. They have been shown to affect the size of lipid droplets and, therefore, fat supplies [14]. Triple ATL knockout affects the differentiation of NIH cells into fat cells, and it also affects the bone morphogenetic protein (BMP) pathway, making the cell more sensitive to ER stress [14]. ATL2 and ATL3 participate in the first step of autophagy, where they direct the ULK1 protein complex to the ER [15]. They also have a role in macro-ER-phagy, which is part of the maintenance of the ER. ATL2 has been shown to maintain ER homeostasis in healthy mouse mammary epithelial cells, so that lipid droplets of the right size and composition can be produced and secreted into breast milk [16]. Herein, its expression was found to be regulated by miR-30b-5p, a member of the miR30 family known to be important for adipogenesis [17]. MiR-30e-5p was found to downregulate ATL2 in the synovial tissue of a mouse model with rheumatoid arthritis [18].

ATL1 is mostly expressed in the central nervous system (CNS), whereas ATL2 and ATL3 have a ubiquitous expression pattern. There are known mutations in ATL1 and ATL3 that result in hereditary spastic paraplegia [19,20], and a role for the ATLs in Alzheimer’s disease (AD) has been suggested as well. Mutations found in the Presenilin-1 (PS1) gene in familial AD have been analyzed in knock-in mouse models [21]. Hippocampal differential gene expression analysis in the PS1 M146V knock-in mouse showed an increase in ATL2 mRNA expression and an increase in contact between the ER and mitochondria, resulting in a change in membrane potential, ROS, and superoxide production. ATL2 was also increased in the brains of 3xTg-AD mice and AD patients [21].

The association of ATL2 with BC has not been determined previously. Herein, we addressed the relationship of ATL2 with BC. BC is the most common cancer in women worldwide [22]. Treatment depends on the expression of the estrogen, progesterone, and/or HER2 receptors [23,24]. It can also be guided by the molecular subtypes basal-like, HER2-positive, and luminal A and B that are based on the tumors’ gene expression patterns [25,26]. In general, the prognosis for BC patients is good, but it depends on the type of tumor, metastasis, and the patient’s response to the treatment. Exploring changes in genetic status, particularly expression, and how it relates to the progression of breast tumors can reveal genes and pathways that may become future drug targets. ATL2 came to our attention during a study of BC families with many affected individuals, a study that is still ongoing. ATL2 is known to be expressed in breast tissue [16], and in one study it was identified among 14 other genes whose expression correlated with metastasis to bone [27]. ATL2 has been associated with other cancer types as well. In non-small-cell lung cancer, it was one of four genes that was identified through a comparison of RNA networks in EGFR wild-type and mutant tumors as a gene that was correlated with shorter survival in lung adenocarcinoma (LUAD) [28]. ATL2 exons 3 and 4 are host sequences to a circular RNA, Hsa-circ 0000993, whose overexpression in gastric cancer cells resulted in decreased migration, invasion, and proliferation but without affecting ATL2 expression [29]. In salivary gland neoplasms, ATL2-PRKD3 fusion genes were identified [30]. PRKD3 is a serine/threonine kinase that affects diacylglycerol (DAG) and phorbol esters, vesicle trafficking, and growth regulation in BC and prostate cancer (PC). In addition, our study demonstrates that a high expression of ATL2-2, a particular ATL2 gene product, is associated with BC prognosis, as well as pathways that support BC development.

## 2. Materials and Methods

### 2.1. Cohorts and Clinical Data

Cohort 1 and Cohort 2 consist of patient data and tumors from 158 and 291 BC patients that were diagnosed in 1987–2003 and 2003–2007, respectively, as previously described [31,32]. Primary fresh frozen breast tumors and adjacent non-neoplastic breast tissue (normal) were obtained from the Department of Pathology, as described [33]. All patients gave informed consent and the National Bioethics Committee of Iceland granted approval for the study (VSN-11-105 and VSN-15-138). Patient and tumor data for METABRIC and TCGA BC cohorts were obtained through cBioPortal [34,35] and from Rueda et al. [36]. BC patients in METABRIC (*n* = 2509) were diagnosed in 1980–2005 [36,37,38] and in TCGA between 1988 and 2013 (Firehose legacy, *n* = 1108) [39].

### 2.2. ATL2-2 mRNA

The ATL2 transcript ENST00000419554.6 encodes 579 amino acids and is referred to as ATL2-2 to keep with the nomenclature in Crosby et al. [40]. Total RNA was isolated from fresh frozen breast tumors and normal breast tissue [31,33]. Quantification of ATL2-2 was performed with quantitative real-time PCR using Prime Time Gene expression assays from IDT with an ATL2-2-specific probe spanning exons 12 and 13a (Hs.PT.24400578, IDT genomics) with TATA-binding protein (TBP, Hs.PT.39858774) as a reference gene. The reactions were performed according to the manufacturer’s protocol, including verification and optimization of the assay. The qPCR reactions were performed in triplicate using 40 cycles and 10 ng of cDNA as a template. ATL2-2 expression was calculated relative to TBP expression: 2^−(mean Ct target − mean Ct reference)^. ATL2-2 mRNA values were obtained from 143 and 265 tumor samples and 36 normal specimens. ATL2-2 mRNA from the METABRIC cohort was retrieved through cBioPortal with measurements available from 1980 tumors. The Illumina gene expression array HT-12 v3.0 was used to quantify METABRIC gene expression. ATL2-2 transcript NM_022374.1 was detected by ILMN_18684, probe located in the 3′ end of the gene. The RNA-Seq ATL2-2 transcript expression data for the BC cohort from TCGA were retrieved through the Xenabrowser portal [41,42]. ATL2-2 mRNA was expressed in 838 tumors of a total of 1085 that expressed various ATL2 transcripts.

### 2.3. ATL2-2 Protein Expression

Tissue microarrays (TMA) from 13 invasive breast tumors and adjacent normal (non-neoplastic) breast tissue were stained with anti-ATL2 antibody (HPA029108, Atlas Antibodies, 0.1 mg/mL, Stockholm, Sweden). Each tumor on the TMA was represented in triplicate and normal tissue was represented in quadruplicates (each core 1 mm in diameter). TMA tissue sections were 5 µm. The antigen retrieval was performed at pH9, and subsequently, the tissue was stained with antibody for 2 h at room temperature at a dilution of 1:20. Visualization was performed with the anti-rabbit EnVision/HRP (Dako, Glostrup, Denmark) according to protocol. The slides were analyzed by two observers, one of whom is a pathologist. The stain in the tumors was mostly cytoplasmic, both granular and diffuse. In the normal breast tissue, there was either no stain or very low levels of a non-specific stain. The scoring system was based on the intensity (none, weak, medium, and strong) and the estimated area of breast cells stained (<5, 5- < 25, 25- < 50, 50–100). The results of the scoring and the scoring scheme are shown Appendix A. The HPA029108 ATL2 antibody, usable only in immunohistochemical applications, detects ATL2-2 and potentially ATL2-3 (see ATL2 transcripts in Appendix A).

### 2.4. Gene Set Enrichment Analysis

Tumors in the whole METABRIC cohort or in the estrogen-receptor-positive luminal B subgroups of METABRIC and TCGA 2012 cohorts were divided into two groups based on high (above median) or low (below median) ATL2-2 mRNA levels. Differences in gene expression (DGE) between the high- and low-expressing tumors were analyzed in the cBioPortal [34,35]. The results were exported and used to perform a pre-ranked analysis in GSEA [43], version 4.3.2, using Hallmark pathways from MSigDB v2023.1 [44,45]. The threshold for significance was set at 5% FDR (false discovery rate). The datasets in these cohorts are quantified on microarrays. The Illumina array HT-12 v3.0 used by METABRIC detected ATL2-2 mRNA, but the Agilent G4502A array used by TCGA detected ATL2-2 (A_23_P415015) as well as ATL2-1 (A_23_P209619). To identify microRNAs associated with high ATL2-2 mRNA levels, the genes that were significantly upregulated when analyzing DGE in estrogen-receptor-positive luminal B tumors from METABRIC were analyzed with the Enrichr algorithm [46,47,48,49].

### 2.5. Statistical Analyses

Patients that lacked ATL2-2 mRNA and breast-cancer-specific survival (BCSS) data were excluded from the analyses. The number of patients in each cohort were 135, 263, 820, and 1902 in Cohort 1, Cohort 2, TCGA, and METABRIC, respectively. The mRNA values from Cohort 1 and Cohort 2 were transformed by log2 to normalize the data; the values from TCGA and METABRIC had been normalized already. ATL2-2 mRNA values in all cohorts were centered at 0. The statistical program R version 4.3.0 was used for the analyses [50]. The ATL2-2 mRNA levels in breast tumors and normal breast tissue were compared using a Student’s *t*-test with unequal variance or a paired *t*-test with unequal variance, confidence level 0.95. Association of ATL2-2 mRNA with clinical and pathological variables was analyzed using a Student’s *t*-test or ANOVA. Kaplan–Meier and log rank tests were calculated to estimate BCSS using the survival and survminer packages in R. Tumors were classified as high- or low-ATL2-2-mRNA-expressing tumors using the maxstat test in R, except in the analysis shown in Appendix A where the division of tumors was based on median ATL2-2 mRNA levels. Cox regression analyses were performed to calculate the hazard ratio (HR) and the effect of tumor characteristics, such as estrogen receptor status, on survival. Tumor characteristics with numerical values were analyzed as both categorical and continuous variables. P-values below 0.05 were considered significant.

## 3. Results

### 3.1. High Expression of ATL2-2 in Breast Tumors

The ATL2 transcripts found in breast tumors from TCGA were analyzed in the Xenabrowser database [41]. Four of seventeen transcripts were expressed at high levels (mean TPM log2 > 0) and in most tissue samples, tumor and normal (Appendix A). However, comparing the four ATL2 transcripts between tumor and normal, only the ATL2 transcript ENST00000419554.6 was more highly expressed in breast tumors than in normal breast. This particular transcript encodes a protein of 579 amino acids (referred to as ATL2-2 in accordance with Crosby et al. [40]). On the other hand, the transcript encoding the longest isoform, ENST00000378954.8, that expresses 583 amino acids (ATL2-1) was equally expressed in breast tumors and normal breast tissue. The third transcript, ENST00000406122.5, encodes a protein of 413 amino acids (ATL2-3) that lacks the N-terminus and a large part of the GTPase domain. Finally, the fourth transcript (ENST00000477642.5) appears to be incomplete as it consists of three exons with an open reading frame, but it lacks a promoter and 3′ UTR. We followed up on the ATL2-2 transcript because it was the only ATL2 transcript with higher expression in breast tumors than in normal tissue.

ATL2-2 mRNA was measured in breast tumors and in adjacent normal breast tissue from BC patients in Cohort 2. ATL2-2 mRNA was expressed at higher levels in breast tumors (n = 263) than in normal breast tissue (n = 36) (*p* = 6.0 × 10^−6^, Figure 1A). Additionally, ATL2-2 mRNA levels were higher in breast tumors than in normal breast tissue in the same individual (paired *t*-test, *p* = 0.05, Figure 1B). Furthermore, ATL2 protein expression was assessed in 13 patient breast tumor samples and matched normal breast tissue with a C-terminus-specific antibody that detects the expression of the ATL2-2 isoform and potentially ATL2-3 (Figure 1C, Appendix A). The results confirmed higher expression of ATL2 in tumors than in normal tissue, likely due to the expression of ATL2-2 since that is the only ATL2-2 transcript with increased expression in tumors compared to normal breast tissue, as stated earlier. We therefore asked if the high expression of ATL2-2 in tumors correlated with more aggressive pathological and clinical characteristics.

### 3.2. High Expression of ATL2-2 Associated with Parameters That Indicate Worse Prognosis

Association analyses between ATL2-2 mRNA and clinicopathological characteristics were performed in BC Cohort 1 (n = 135), Cohort 2 (n = 263), TCGA (n = 820), and METABRIC (n = 1902) (Appendix A, respectively). We found that ATL2-2 mRNA expression was higher in estrogen-receptor-negative tumors than in estrogen-receptor-positive tumors in all four cohorts. That data reached statistical significance (*p* < 0.05) in Cohort 2 and METABRIC (Figure 2 and Appendix A). A significantly higher expression of ATL2-2 mRNA was observed in large tumors (>20 mm) as compared to small tumors (≤20 mm) in Cohort 2 and METABRIC and in grade 3 tumors as compared to lower grade tumors in Cohort 1, Cohort 2, and METABRIC (Appendix A, *p* < 0.05). The size and grade are incorporated into tumor stage in TCGA and, thus, these variables were not analyzed individually in this cohort. Not surprisingly, the highest ATL2-2 mRNA expression was observed in the basal molecular subtype, which includes mostly estrogen-receptor-negative tumors, although there was high expression in some luminal B and claudin-low tumors (Appendix A). Taken together, ATL2-2 expression was higher in tumors that were estrogen-receptor-negative, large (>20 mm), grade 3, and basal-like, all factors that indicate a more aggressive disease.

### 3.3. High ATL2-2 mRNA Levels Associated with Shorter Breast-Cancer-Specific Survival

ATL2-2 mRNA levels did not associate with survival in any of the cohorts (as an example, see results for METABRIC in Appendix A). This was not unexpected due to the association of ATL2-2 mRNA with estrogen receptor status. Therefore, survival was analyzed separately in patients whose tumors were either estrogen-receptor-negative or -positive. Most of the estrogen-receptor-negative tumors are classified as basal-like tumors according to the molecular subtype, and estrogen-receptor-positive tumors are mostly of the luminal subtype, luminal A and B. Since the METABRIC cohort was the largest of the four cohorts in this study, it was chosen to analyze breast-cancer-specific survival (BCSS) according to estrogen receptor and molecular subtype status. ATL2-2 mRNA levels did not associate with survival in the estrogen-receptor-negative basal-like subgroup in METABRIC. However, in the estrogen-receptor-positive luminal subgroup, there was an association of ATL2-2 expression levels with shorter BCSS (Figure 3). The hazard ratio (HR) was 1.535 (CI 1.228–1.918, *p* = 2 × 10^−4^). The clinicopathological analyses revealed that the strongest association was between ATL2-2 mRNA levels and tumor size, grade, and molecular subtype. Thus, not surprisingly, these variables all attenuated the association of ATL2-2 with BCSS (Table 1). Nonetheless, after taking the effect of these clinicopathological characteristics into account, HR remained significant. The HR was 1.334 (CI 1.063–1.673, *p* = 0.013) after correcting for the molecular subtype, which was the strongest confounder. High ATL2-2 mRNA levels associated with shorter BCSS in patients with estrogen-receptor-positive luminal tumors in Cohort 1. They also associated with BCSS in TCGA patients who were diagnosed in the period between 1988 and 2005 (Appendix A), which is the same diagnostic period as that of the METABRIC cohort. Molecular subtyping has not been determined for tumors in Cohort 2. Furthermore, even when analyzed separately in estrogen-receptor-positive luminal B tumors from METABRIC, high ATL2-2 mRNA levels associated with BCSS (Appendix A) after correction for grade (HR 1.478, CI 1.059–2.064) and size (HR 1.459 CI 1.047–2.033) (Appendix A). In summary, ATL2-2 mRNA levels associated with shorter BCSS in patients with estrogen-receptor-positive luminal tumors.

### 3.4. High ATL2-2 mRNA Associated with Proliferative Pathways

To shed light on the potential role of ATL2-2 in cancer progression, gene set enrichment analysis was performed in the GSEA software to examine whether high ATL2-2 mRNA levels associated with known biological pathways. In the whole METABRIC cohort, 21 Hallmark pathways were significantly enriched (*p* < 0.05 and FDR cutoff 5%) when ATL2-2 expression was high (Appendix A). These pathways can be placed into three categories: immunology, proliferation, and vesicle-related functions. To narrow the search for cancer-related pathways, the analysis was repeated in estrogen-receptor-positive luminal B tumors from METABRIC and TCGA cohorts. The Hallmark pathways that were significantly enriched in high-ATL2-2-expressing tumors from both cohorts were MYC targets v1, E2F targets, and the G2M checkpoint, the same pathways that had the highest enrichment score (ES) in the analysis of the whole METABRIC cohort (Figure 4, Appendix A, METABRIC and TCGA, respectively). These three pathways are known to be upregulated in tumors and all three induce proliferation, which in turn supports tumor progression. To analyze whether upregulated genes in ER-positive luminal B tumors that express high ATL2-2 mRNA associate with microRNAs, the genes were analyzed in Enrichr. The hsa-miR30 gene family was the most highly associated miRNA (*p* = 1 × 10^−6^, Appendix A). There is a known relationship between ATL2 and miR30, and therefore, we consider our finding herein significant [16].

## 4. Discussion

Herein, we found that the ATL2-2 transcript and protein were higher in breast tumors than in normal breast tissue. High ATL2-2 mRNA associated with estrogen-receptor-negative, basal-like, high-grade, and large tumors, all of which signal a worse prognosis. Also, high ATL2-2 mRNA associated with shorter BCSS and the upregulated expression of genes in proliferative pathways. Thus, our study suggests that the high expression of ATL2-2 supports tumor progression in BC cells.

The increased expression of ATL2-2 may serve as a regulatory mechanism to promote cancer cell proliferation and metastasis. Seventeen ATL2 transcripts were identified in TCGA breast tumors in the Xenabrowser database, and it is likely that some of them are incomplete transcripts due to the lack of 5′ and 3′ UTRs. Five ATL2 transcripts were reported by Tapial et al. [51], and six ATL2 transcripts can be found in the UCSC browser [52]. The variation among these is observed in both the 5′and 3′ ends of the gene. Interestingly, ATL2-1, the longest transcript, was expressed in all TCGA tumors as examined in the Xenabrowser database, whereas ATL2-2 was expressed in 75% of tumors. A recent biochemical study that focused on the ATL2-1 variant demonstrated that the C-terminus contained an inhibitory domain, which must be released before the protein can function as a GTPase and join ER membranes [40]. It is not known what releases the inhibition. Also shown was that the ATL2-2 variant has a different C-terminus that lacks the amino acids that inhibit the GTPase activity and that the uncontrolled production of ATL2-2 resulted in a collapse of the ER [40]. ATL2 functions as a dimer, and this suggests that a balance between the ATL2 isoforms is needed to maintain ER homeostasis. The overexpression of ATL2-2 and the collapse of the ER may be useful at some points in the life of a cell, perhaps during the shape changes that occur during the division, migration, and involution of organs and for the maintenance of the ER.

The pathological variables of estrogen-receptor-negative tumors, high grade and large size, are all independent prognostic factors that are associated with survival. High ATL2-2 mRNA did not associate with survival in the estrogen-receptor-negative group, perhaps not surprisingly as a negative estrogen receptor status is a very strong prognostic factor, and the effect of ATL2-2 is too weak to be observed in this group. Thus, it was more likely to see an effect in the estrogen-receptor-positive group. Large tumors and high-grade tumors can be either estrogen-receptor-positive or -negative and indicate a worse prognosis regardless of estrogen receptor status. However, even after taking tumor size, grade, and molecular subtype into account in the survival analysis, high ATL2 mRNA remained associated with BCSS in the estrogen-receptor-positive luminal group in the METABRIC cohort. The METABRIC cohort was used for the survival analysis because it was the largest cohort which allowed for a reasonable number of patients in the groups after division according to estrogen receptor status and molecular subtype. In Cohort 1, there was a trend towards the association of high ATL2-2 with shorter BCSS, but it was not significant. The tumors from Cohort 2 have not been analyzed for molecular subtype, and thus, survival analysis according to estrogen receptor status and molecular subtype was not performed. ATL2-2 was not associated with BCSS in TCGA, unless patients diagnosed during the years 1988–2005 were analyzed separately. Unlike the other cohorts, the majority (79%) of the TCGA cohort was diagnosed in 2006 and later. The cut-off of 2005/2006 was chosen to mimic the diagnostic years in METABRIC. The discrepancy in survival analyses between cohorts may be due to different treatments that the patients received as they were collected into the studies over long periods, and patient composition may be different as well.

GSEA showed that genes that were upregulated in breast tumors with high ATL2-2 mRNA levels correlated with genes in the Hallmark pathways E2F targets, MYC targets, and G2M checkpoint. These three pathways are known to be active in cancer biology and they support proliferation, which is a hallmark of cancer biology. It is possible that high ATL2-2 expression supports normal cellular proliferation that is hijacked by cancer-inducing mechanisms. MYC and E2F1 are both transcription factors but whether they induce the expression of ATL2-2 is unknown. Based on the reported ATL2 functions, there are numerous ways that ATL2 could support cell proliferation and potential tumor progression. ATL2 participates in the initial steps of autophagy [15]. Although ER-phagy is required for the maintenance of the ER (reviewed in [53]), the autophagy process is used to produce macromolecules in hypoxic environments, such as is seen in a growing tumor before it metastasizes (reviewed in [54]). ATL2 could also support tumor progression by affecting proteins in the ER (FAM134B). Calcium storage and release from the ER is tightly controlled by calcium channels. As the most important second messenger in the cell, calcium controls many pathways in both healthy and malignant cells [55]. The IP_3_R1-3 family of ER calcium channels control many of these pathways. ATL2 has been shown to induce IP_3_R calcium release. It occurs when ATL2 reshapes the ER, which redistributes the RyR2 calcium channel and calcium release, inducing IP_3_R calcium release [56]. In comparison to ATL1 in which mutations result in spastic paraplegia, which emphasizes the importance of a branched ER network in neuronal axons ([57] and references therein), ATL2 may be important in filipods and other mechanisms involved in cellular migration.

ATL2 is likely of importance for a healthy ER. Enhanced expression of the ATL2-2 variant may offset the balance in the ER, making the cytoplasmic environment more fit to support cancer mechanisms. The results presented herein show that ATL2-2 was more highly expressed in breast tumors than in normal breast tissue, high ATL2-2 mRNA levels are associated with clinicopathological parameters that can worsen prognosis and shorten breast-cancer-specific survival and the upregulation of genes that can control proliferative pathways. Taken together, the results suggest that high expression of ATL2-2 may support tumor progression.

## Figures and Tables

**Figure 1 genes-14-01559-f001:**
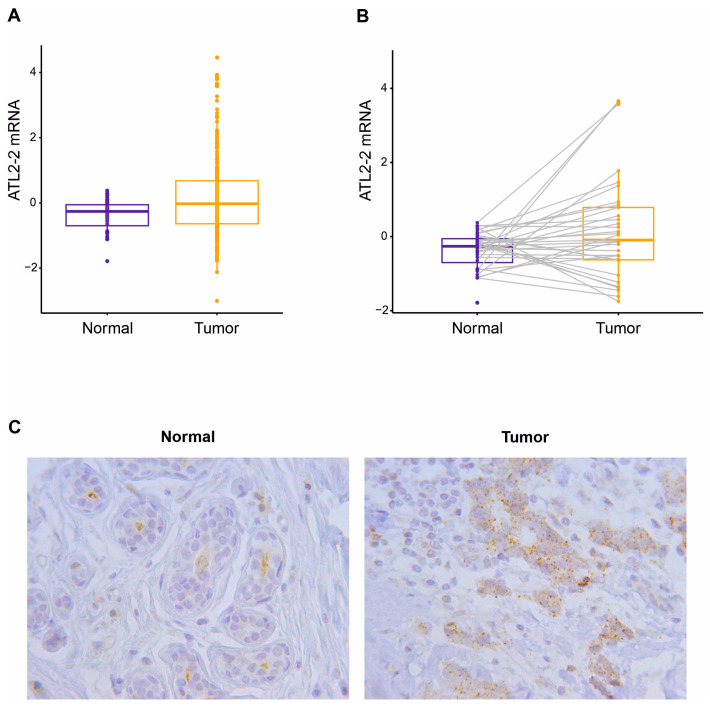
ATL2-2 is more highly expressed in breast tumors than normal breast tissue. (**A**) ATL2-2 transcript levels were compared between breast tumors and normal (non-neoplastic) breast tissue samples from Cohort 2 using a Student’s *t*-test (263 tumors vs. 36 normal tissue), *p* = 6.0 × 10^−6^, and (**B**) in 36 tumors and their corresponding adjacent normal tissue using a paired *t*-test, *p* = 0.05. Tumors are depicted in yellow and normal tissue in blue. (**C**) ATL2 protein expression was detected by ATL2 antibody in tumor cells (granular brown cytoplasmic stain) and adjacent normal cells (faint brown). A representative figure from immunohistochemical analysis of 13 tumor–normal pairs. The magnification was 40x. ATL2 protein expression was mostly observed in the cytoplasm in granules and as a diffuse stain.

**Figure 2 genes-14-01559-f002:**
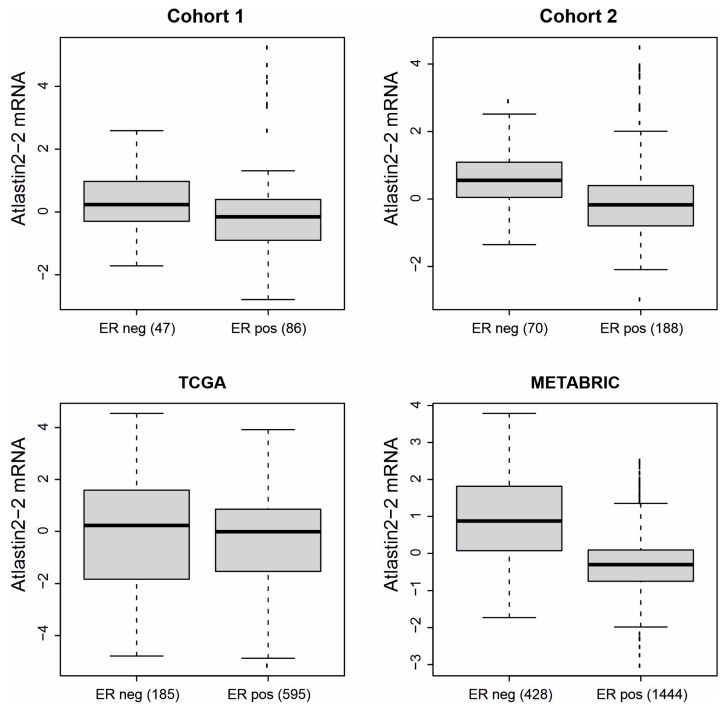
ATL2-2 mRNA expression levels were higher in estrogen-receptor-negative tumors as compared to estrogen-receptor-positive tumors. A correlation analysis between ATL2-2 mRNA levels and estrogen receptor status was performed in Cohort 1, Cohort 2, TCGA, and METABRIC. The number of estrogen-receptor-negative and -positive tumors are shown below each boxplot. The difference in ATL2-2 expression between the estrogen-receptor-negative and -positive categories was calculated with a *t*-test with unequal variance. Cohort 1: *p* = 0.089, Cohort 2: *p* = 4 × 10^−5^, TCGA: *p* = 0.083, METABRIC: *p* = 2.2 × 10^−16^.

**Figure 3 genes-14-01559-f003:**
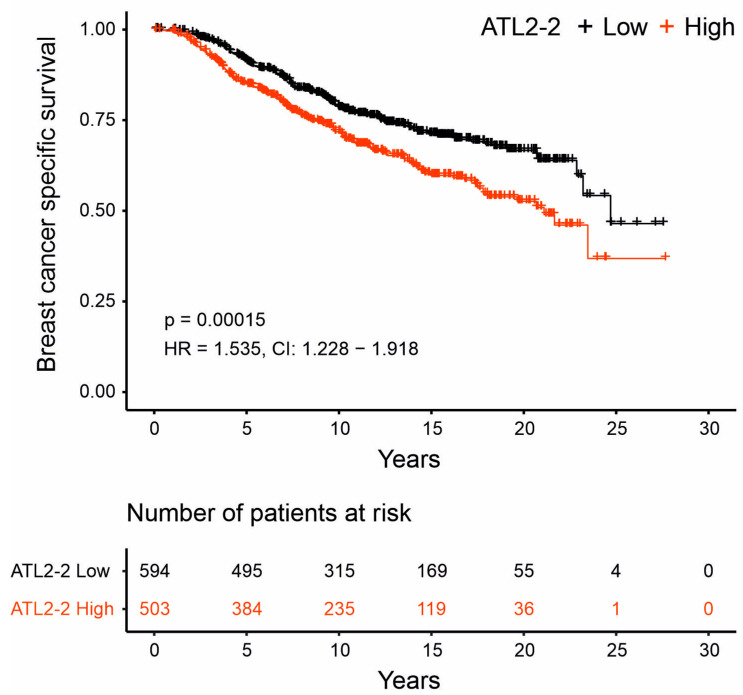
High ATL2-2 mRNA levels associated with shorter breast-cancer-specific survival in patients with estrogen-receptor-positive luminal tumors. Breast-cancer-specific survival (BCSS) was analyzed in the METABRIC cohort in patients whose tumors expressed the estrogen receptor and were classified as luminal according to molecular subtyping. The ATL2-2 mRNA values in the patients’ tumors were divided based on the max stat function that finds the best division based on outcome. In the low-expressing group, there were 594 (black line), and there were 503 in the group expressing high ATL2-2 (red line). The log rank p-value was 2 × 10^−4^. The number of patients at risk at the indicated time point is shown in a table below the Kaplan–Meier graph. The HR was 1.535 (CI 1.228–1.918) prior to adjusting for confounding variables. Table 1 shows the hazard ratios (HRs) and confidence interval (CI) from the Cox regression analysis prior to and after adjusting for confounding variables.

**Figure 4 genes-14-01559-f004:**
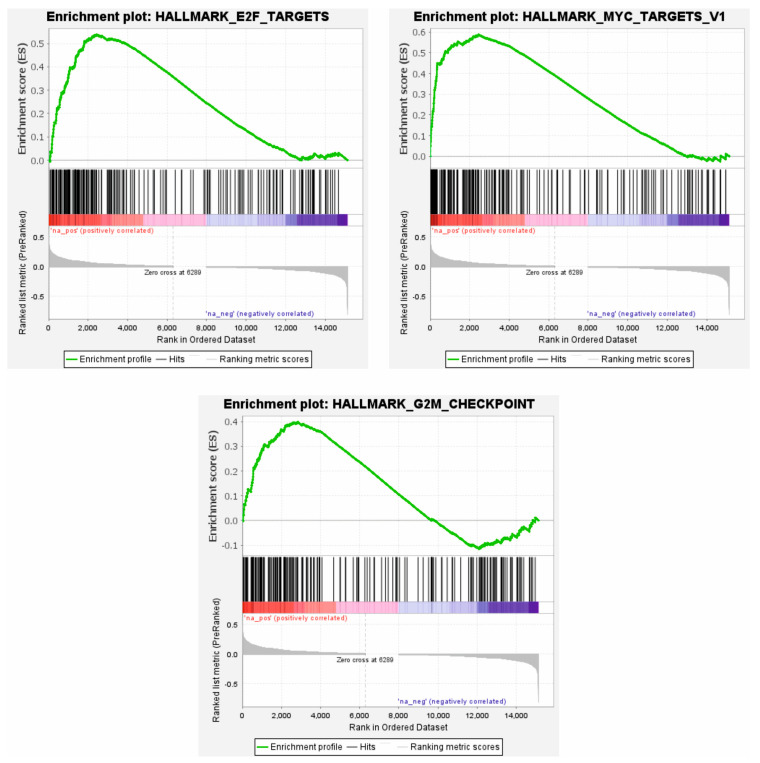
Proliferative pathways were enriched in tumors that express high ATL2-2 mRNA levels. Gene set enrichment analysis was used to analyze which Hallmark pathways were upregulated in tumors with high ATL2-2 mRNA levels. The results show the top three pathways identified in the whole METABRIC cohort, which were also the only three pathways observed in estrogen-receptor-positive luminal B tumors in both METABRIC and TCGA cohorts.

**Table 1 genes-14-01559-t001:** Adjustment of BCSS for confounding characteristics in patients from METABRIC with estrogen-receptor-positive luminal tumors (n = 1097).

	HR	CI	*p*-Value
High ATL2-2	1.535	1.228–1.918	1.6 × 10^−4^
+age at diagnosis ^1^	1.550	1.240–1.937	1.2 × 10^−4^
+grade	1.429	1.139–1.794	2.1 × 10^−3^
+tumor size ^1^	1.436	1.147–1.799	1.6 × 10^−3^
+nodes	1.559	1.248–1.949	9.4 × 10^−5^
+stage	1.473	1.144–1.898	2.7 × 10^−3^
+molecular subtype	1.334	1.063–1.673	1.3 × 10^−2^
+HER2	1.512	1.210–1.890	2.8 × 10^−4^
+Progesterone receptor	1.480	1.183–1.851	6.0 × 10^−4^

^1^ Continuous variables.

## Data Availability

Publicly available datasets were analyzed in this study. METABRIC gene expression data and patient data were retrieved from the cBioPortal: https://www.cbioportal.org/ (accessed 8 February 2022 for gene expression data and 6 October 2021 for clinical data) under “invasive breast carcinoma”. RNA-Seq data from TCGA were retrieved from https://xenabrowser.net/ (accessed on 26 May 2021). Patient data generated by the TCGA research network were retrieved from the cBioPortal: https://www.cbioportal.org (accessed on 20 November 2020) under “invasive breast carcinoma”.

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
