# Peer review of "High Atlastin 2-2 (ATL2-2) Expression Associates with Worse Prognosis in Estrogen-Receptor-Positive Breast Cancer"

_genes, 2023, doi:10.3390/genes14081559_

Round 1
Reviewer 1 Report
In this manuscript Reynisdottir et. al. report an association of Atlastin 2 isoform 2 (ALT2-2) in breast cancer prognosis. TGCA and METABRIC data sets were analyzed for the expression of different ALT2 transcript isoforms which indicated selective upregulation of ALT2-2 in BC compared to normal breast tissue. This observation was further validated in independent cohorts of breast cancer samples. Further on ALT2-2 expression was found to be higher in estrogen receptor (ER) negative BC compared to ER+ BC. However, ATL2-2 mRNA levels did not predict a prognosis value while it is associated with shorter BCSS in patients with ER+ luminal tumors. Overall, study was designed and executed well, and the Introduction, Results and Discussion sections was written clearly and coherently. However, some statements are overinterpretation and needs to be toned down. For instance, Line 47 “Taken together, the results indicate that high levels of ATL2-2 are necessary to support BC progression.” Authors do not provide any evidence for the necessity of ALT2-2 in BC progression. All they have shown is association.
Additional major comments:
1. Figure 1C: How do authors determine tumor tissue and adjacent normal cells in the same TMA section? Co-staining (or staining in adjacent section) with a tumor specific marker should be performed in those TMA to determine tumor versus nontumor cells for the analysis of ATL2-2 protein expression.
2. Twelve tumor normal pairs were stained whereas only one representative image was shown. At least one normal and one tumor should be shown and quantification of the staining for all the sample should be presented.
3. Authors should include a positive control for the specificity of the antibody to ATL2-2 and show that it does not recognize other isoforms. This could be done by a western blot where the isoforms could be determined by their molecular weight.
4. Enlarge the figure labeling for clarity.
Author Response
Response to reviewer #1
The authors would like to thank the reviewer for the valuable comments.
We appreciate the point that our study shows association, but it does not prove that ATL2-2 is necessary for breast cancer progression. Therefore, we have toned down our statements in the following places:
Lines 52-53, 120-122, and 348-349 and in the graphical abstract.
Response to major comments:
- In our Department of Pathology breast tissue is routinely stained with H&E (hematoxylin and eosin stain) in order to diagnose if cancer is present. As in other pathology departments, this is done using standard histological criteria (pleomorphism, pattern of growth, loss of differentiation, invasiveness etc.) rather than using a tumor specific marker and we are in fact not aware of any marker which would be specific enough to make this determination. In our case the diagnosis of breast cancer was made by one of the authors who is a pathologist with much experience in breast pathology. Subsequently, after a diagnosis of cancer has been made as outlined above, the tissue is analyzed using antibodies to the estrogen (ER) and progesterone (PgR) receptors, HER-2, and Ki-67 in order to classify the cancer further and assist with treatment decisions.
- We added a new Figure 1C with a tumor-normal pair from the same patient. The scoring scheme and the final score for all the pairs can now be found in a new supplemental Table, Table S1. The Methods section on immunohistochemistry has been expanded to include a more detailed description of the procedure and the scoring (lines 163 – 169) and changes made in lines 226-233 in Results.
- A very good point that we agree with. The HPA029108 antibody that was used in the study only works in immunohistochemistry, but not in western blot, and thus we cannot verify that it detects only ATL2-2. The immunogenic sequence is derived from the unique C-terminus of ATL2-2. If the antibody detects another ATL2 isoform it should only be ATL2-3 because it is the only other isoform that has sequence similarity to ATL2-2 in the C-terminus (see transcripts in Figure S1). The two isoforms have an identical C-terminus with the inclusion of five amino acids in ATL2-3. We do not know whether these extra ATL2-3 amino acids interfere with the detection of this isoform because the antibody does not work on a western blot. We have included this information in the manuscript in Methods (lines 163-169).
- The labels in the figures have been enlarged.
Reviewer 2 Report
In this manuscript, the authors use bioinformatic methods to find that ATL2 associates with worse prognosis in estrogen receptor positive breast cancer. Atlastin 1-3 (ATL1-3) are a family of membrane bound GTPase proteins located in the ER. It is of interest and my suggestions are as follows,
1. Why the author choose ATL2 instead of other two genes to analyze?
2. It is necessary to add experimental validations such as RT-qPCR, WB, etc to verify their findings.
3. More bioinformatic analyses should be added at single-cell or spatial RNA seq level.
4. The font sizes are too small, which should be adjusted.
5. In Figure 3, what does the "Time" mean? Besides, the Hazard ratio and confidence interval values should be added.
6. The introduction and discussion part should be more detailed. For example, there are numerous studies focused on breast cancer (PMID: 36759097, 35499382, 37068524, etc.).
7. The manuscript should be edited by a native speaker.
None
Author Response
Response to reviewer #2
The authors would like to thank the reviewer for the valuable comments.
1. The main research focus of the authors is breast cancer. We chose to study ATL2 rather than ATL1 or ATL3 because ATL2 had previously been shown to be expressed in murine breast tissue and it had been associated with cancer in a few studies, including in breast cancer. We have also seen a genetic variant in the gene in a breast cancer family in an ongoing study. We have tried to clarify this by adding sentences that can be found in lines 105-109.
2. We agree with the reviewer that experimental validations are necessary. We do not have access to samples from METABRIC and TCGA and thus we performed validation studies in our own cohorts. RT-qPCR was performed to quantify ATL2-2 in Cohorts 1 and 2 to validate the clinical and pathological findings in METABRIC. Immunohistochemistry (IHC) with an ATL2 antibody was performed in breast tumors and normal breast tissue from breast cancer patients to confirm the results from the mRNA measurements. The antibody did not work for western blots. IHC was also performed on breast tissue microarrays from Cohort-1. However, we could not use it to replicate survival analysis because the number of samples becomes too small when only ER-positive luminal tumors are used.
3. This is a great idea that will be worth pursuing when the study is more advanced. However, we believe that this is beyond the scope of the current study.
4. The font sizes in the figures have been increased.
5. In Figure 3, the label “Time” on the x-axis refers to years until the patient dies from breast cancer. It is “Time (years)” on the x-axis and “Breast cancer specific survival” on the y-axis. We changed the label on the x-axis to “Years” for clarification. Hazard ratio and confidence interval have been added to Figure 3.
6. Thank you for pointing out these references to us. Our understanding of this comment is that we should include more information about breast cancer. We have included some background information on breast cancer in lines 99-105. To stay within the limits of the journal, we kept it short but included some of the suggested references.
7. The manuscript has now been read and edited by a native English speaker Dr. Jill Bargonetti, who is a Professor in Biology at Hunter College in New York City. Her research focuses on breast cancer. The changes in the manuscript that are not responses to the reviewers´ comments are due to her editing.
Round 2
Reviewer 1 Report
The revised manuscript addressed all my comments.